# Effect of intra- and inter-specific plant interactions on the rhizosphere microbiome of a single target plant at different densities

Derek R. Newberger[1], Heather L. Deel[2], Daniel K. Manter[2], Jorge M. Vivanco[1]*

1 Department of Horticulture and Landscape Architecture and Center for Rhizosphere Biology, Colorado State University, Fort Collins, Colorado, United States of America, 2 Soil Management and Sugar Beet Research Unit, United States Department of Agriculture Agricultural Research Services, Fort Collins, Colorado, United States of America

* J.Vivanco@ColoState.edu

## Abstract

Root and rhizosphere studies often focus on analyzing single-plant microbiomes, with the literature containing minimum empirical information about the shared rhizosphere microbiome of multiple plants. Here, the rhizosphere of individual plants was analyzed in a microcosm study containing different combinations and densities (1–3 plants, 24 plants, and 48 plants) of cover crops: *Medicago sativa*, *Brassica* sp., and *Fescue* sp. Rhizobacterial beta diversity was reduced by increasing plant density for all plant mixtures. Interestingly, plant density had a significant influence over beta diversity while plant diversity was found to be a less important factor since it did not have a significant change. Regardless of plant neighbor identity or density, a low number of rhizobacteria were strongly associated with each target species. Nonetheless, a few bacterial taxa were shown to have conditional associations such as being enriched within only high plant densities, which may alleviate plant competition between these species. Also, we found evidence of bacterial sharing of nitrogen fixers from alfalfa to fescue. Although rhizosphere bacterial networks had overlapping bacterial modules, the modules showing the largest percentage of the network changed depending on plant neighbor. In summary, this study found that for the most part plants maintained their rhizosphere microbiome despite escalating plant-plant competition.

## Introduction

Roots and soil microorganisms have co-evolved for millions of years [1]. Some root-recruited microorganisms are plant species-specific and provide services that are critical for plant survival [2–4]. Due to these services, the associated microbial community becomes plant-inseparable, collectively forming a holobiont [4].

Although plants and microorganisms may compete for nutrients [5], there is a mutualistic component to their association [6]. Microorganisms have been shown to improve plant fitness by removing environmental stress, moderating plant development, mediating immune responses toward pathogens, and even indirectly influencing plant phenotypic plasticity [1,2]. In exchange, plant roots secrete sugars, organic acids, phenolics, and amino acids, which

**Data availability statement:** Sequencing datasets, custom functions, R Markdown, and metadata generated during and/or analyzed during the current study with R markdown files are on GitHub, [https://github.com/Derek-Newberger/Plant_Neighbor_Rhizosphere.git] and [https://github.com/DanielManter-USDA/DRN-2381389]. Raw data will be granted upon request.

**Funding:** All funding for this original research article came from the National Institute of Food and Agriculture (NIFA)/United States Department of Agriculture (USDA) through a Western Sustainable Agriculture Research and Education (SARE) project [#SW20-910] awarded to JMV. The funders had no role in study design, data collection and analysis, decision to publish, or preparation of the manuscript.

**Competing interests:** The authors have declared that no competing interests exist.

microorganisms utilize as substrates or signals [7,8]. Additionally, roots can change the soil physical and chemical structure by absorbing moisture, secreting phytochemicals, and sloughing off root cells, while senesced above-ground plant material increases organic carbon in the soil [5,9,10]. Through all of these collective processes plants have a significant influence on microorganism community structure [5] and function [1].

The bulk soil surrounding plants serves as a reservoir of different microorganisms for roots to recruit and culture based on the plant's developmental and environmental needs [11–14]. Different root exudates and architectures attract distinct microorganisms to plants [15,16]. After being drawn to the roots by exudates, microorganisms take residence in the rhizosphere, a narrow region where the roots interact with the soil [10,17].

In nature, plants are often surrounded by several neighboring plants of either the same or different species. The strategies of coexistence in plant-plant interactions may be partially founded on tolerating the microorganisms recruited by neighboring plant species [18]. According to the competitive exclusion principle, niche redundancy between microorganisms will eventually lead to some being outcompeted and consequently lost within the rhizosphere of different plant species [19,20]. Thus, microbial colonization in the shared rhizosphere of multiple plant species may follow a "first come, first served" strategy [19]. Plants use multiple strategies to modulate their rhizosphere colonization, such as suppression or induction through secretion of secondary metabolites, competitive exclusion, induced resistance, or an amalgamation of these strategies [19]. Similar to how the foundation of the human microbiome is established as the infant passes through the birth canal [21], the seed represents an important reservoir of plant-beneficial endophytic and epiphytic microorganisms inherited by progeny plants [22–24]. Just as human cohabitation (especially between couples) has been shown to promote shared skin microbiota [25,26], plants grown in proximity may also share microbiota not typically found when grown separately. However, further study is required to view how the colonization of microorganisms in the shared rhizosphere is influenced by plant-plant competition.

Our previous study explored the effect of the density and diversity of cover crops on the bulk soil microbiome and plant biomass [27]. Increasing plant competition was confirmed to decrease individual plant biomass, even if the increase in the number of plants led to a total increase in plant biomass per pot. Additionally, the bulk soil showed a tradeoff of microbes. Differential abundances identified a different group of bacteria for each plant mixture. Here, we used that same greenhouse experimental set up to study the effects plant density and diversity could have on plant bacterial recruitment by investigating the composition of an individual plant's rhizosphere. We utilized a soil that was disinfected via autoclave to amplify the plant effects on rhizosphere microbial composition. This study used alfalfa (*Medicago sativa*), fescue (*Fescue* spp.), and mustard (*Brassica juncea*) to evaluate how interspecific and intraspecific plant-plant interactions modify the composition and functionality of the shared bacterial rhizosphere. These three plants have different growth strategies in terms of leave-shoot development, root architecture, and competitive abilities [28–30], and each recruit distinct microbial species in the root zone [31–36]. This study highlighted a different aspect of plant competition, as it looked at plant bacterial recruitment instead of biomass.

## Methods

### Soil cover crop seed preparation

With the exception of harvesting rhizosphere soil, instead of bulk soil, the methods used in the present study were identical to those in Newberger et al. [2023]. Soil collection occurred at the Agricultural Research, Development and Education Center South, which is owned by Colorado

State University. Metal sieves (2 cm wide) were used to separate large debris from the soil. All soil was pooled prior to autoclaving. The purpose of autoclaving was to augment the plant's impact on the soil microbiome by decreasing soil microbial biomass and microbial community complexity [3,13,37,38]. Approximately 13.5 kg of soil per batch was autoclaved in 61 cm x 76 cm polyethylene autoclave bags using a STERIS steam autoclave (Mentor, Ohio, USA). Soils were autoclaved for three 40-min liquid cycles at 121°C. Soils were then pooled again.

## Density and diversity greenhouse experiment

The greenhouse experiment took place in Colorado State University's Horticultural Center Greenhouse Facility between August 1 to September 1, 2021 (31 days total). This small-scale experiment system efficiently isolated numerous samples by using a microcosm approach. Microcosm was defined as an individual "pot" (6 cm × 4.9 cm × 5.6 cm) from a 36-cell tray, where pots were separated by approximately 2 cm. Microcosms were lined with two layers of medium duty weed fabric (Vigoro Corporation, Lake Forest, Illinois, U.S.A).

There was a total of 21 treatments. A diversity treatment was applied to seven different combinations of alfalfa, brassica, and fescue plants (alfalfa, brassica, fescue, alfalfa-brassica, alfalfa-fescue, brassica-fescue, and alfalfa-brassica-fescue). Seeds used in this greenhouse study were purchased from Vitality (fine fescue species mixture of chewing fescue (*Festuca rubra* ssp. *Commutate*), hard fescue (*Festuca longifolia*), and creeping red fescue (*Festuca rubra*)) and Johnny's selected seeds (ranger alfalfa (*Medicago sativa*) and Mighty Mustard® Pacific Gold (*Brassica juncea*)). For each diversity treatment, there were three density treatments (low: 1–3 total plants per pot, medium: 24 total plants per pot, and high: 48 total plants per pot). Each of the 21 different treatments had 12 replicates for a total of 252 microcosms. Pots were configured into 21 rows by 12 columns using an online random block design generator (<https://www.randomizer.org>). A single plant of each species was also grown in a mesocosm to serve as a control free from inter- and intra-specific competition or facilitation.

Unsterilized plant seeds (to preserve their microbiome) were manually counted for each treatment and dispersed evenly into the pots using autoclaved tweezers. Tweezers were washed with ethyl alcohol in between microcosms. To overcome seed germination failure, unsterilized pregerminated seeds that had been grown in vitro since the start of the greenhouse experiment were planted into each pot seven days into the experiment to reach the target densities. Pots were watered to holding capacity daily. Water was processed using a Synergy® Water Purification Systems (Type 1) by MilliporeSigma (Burlington, MA, USA). This watering technique was employed to reduce the introduction of microorganisms and other substances. Prior to harvesting, the number of plants within each pot was verified.

## Rhizosphere soil collection

Rhizosphere soil samples were collected over four days. Here, rhizosphere soil is defined as the soil that remained adhered to the roots when plants were gently removed from their pot. To ensure that rhizosphere soil was collected from the target plant species, each individual plant was carefully pulled out of the pot leaving behind all bulk soil and overlapping roots. Each plant was separated by plant species per pot, and the attached shoot was used to identify the plant species of each root mass. Five replicates per treatment were collected, of which plant counts that most closely represented the target density and diversity treatment were selected for bacteriome analysis. S1 Table shows the experimental set up of the 21 treatments in the greenhouse, and where the 36 different combinations of rhizosphere samples came from. Roots with rhizosphere soil were placed in 15 ml falcon tubes and immediately stored at −20°C.

## DNA extraction

Rhizosphere soil was removed from plant roots for total genomic DNA (gDNA) extractions. For each sample, 0.25 g of rhizosphere was extracted in a Qiagen QIAcube instrument using Qiagen PowerSoil Pro® DNA kits (Germantown, Maryland, USA) and the manufacturer's protocol. Each DNA extraction was eluted to 100 μl. DNA concentrations were quantified using an Invitrogen Qubit fluorometer (Waltham, Massachusetts, USA) with high sensitivity assay solutions. Rhizosphere samples from each of the 36 treatments had 4-5 replicates. Some samples did not contain enough rhizosphere soil, such that the DNA was not quantifiable after extraction. Pre-extracted Zymo gDNA (Zymo Research Corporation, California, USA) (n = 2) and pre-extracted and sequenced reference soil (n = 2) from Agricultural Research, Development and Education Center-Colorado State University were used as positive controls. Water (ddH$_2$O) was used as a negative control at each step of the pipeline (DNA extraction, PCR1 and PCR2).

## Oxford nanopore sequencing and bioinformatics pipeline

Modifications were made to the 2019 library preparation protocol to sequence full length 16S rRNA gene in Nanopore MinION sequencer http://dx.doi.org/10.17504/protocols.io.[dx.doi.org/10.17504/protocols.io.6j6hcre]. The modified protocol, library preparation and sequencing for MinION – Manter Lab Protocol 2022 was uploaded as a supplementary document (S1 Protocol. Library preparation and sequencing for MinION–Manter Lab Protocol). Qubit concentrations (ng/μl) of DNA were used to calculate a 5x dilution with HPLC water. Primers for bacteriome analysis were Bact_27F-Mn (5' – TTTCTGTTGGTGCTGA TATTGCAGRGTTYGATYMTGGCTCAG – 3') and Bact_1492R-Mn (5' – ACTTGCCT-GTC GCTCTATCTTCTACCTTGTTACGACTT – 3'). A Roche LightCycler® 96 (Basel, Switzerland) was used for PCR. The first PCR settings were 25 cycles of 98°C for 30 sec, 98°C for 15 sec, 50°C for 15 sec, and 72°C for 1 min followed by a single cycle of 72°C for 5 min. Following the initial PCR run, a 1:1 ratio of DNA and beads were combined. Beads with adhering DNA were magnetized to a 96-pronged magnetic stand and rinsed in 70% ethanol for 30 seconds twice. DNA was eluted with 40 μL of PCR grade water, and magnetic beads were removed using a magnetic stand. DNA concentrations were again quantified using Qubit with high sensitivity assay solutions. The second PCR settings were 25 cycles of 98°C for 30 sec, 98°C for 15 sec, 62°C for 15 sec, and 72°C for 1 min followed by a single cycle of 72°C for 5 min. PCR2 was used to add barcodes (EXP-PBC-96). Then following PCR 2, DNA amplicons were cleaned using a bead solution and all samples were pooled into lo-bind 2 ml tubes. A cost-effective method of making the bead solution was used http://dx.doi.org/10.17504/protocols.io.[DOI:10.1101/gr.128124.111] [38]. Samples were then pooled into a single Lo-Bind 2 ml centrifuge tube.

A R9.4.1 flow cell was loaded onto a MinION sequencer. For flow cell preparation, approximately 20 μL of air was drawn out of the flow cell. To prime the flow cell, the priming port was flushed with the buffer solution. Pooled DNA was loaded into the sampling port. The pooled library was sequenced for 48 hours. Guppy v6.0.1 was used to base-call and demultiplex raw data. Sequence reads were filtered by quality and length (Filtlong minimum length: 1000; mean quality: 70; Cutadapt: -m 1000 -M 2000). The EMU NCBI Reference Database was used to identify bacterial taxa. EMU error correction identified and removed bacterial taxa using alignment and abundance profiles, during which bacterial taxa with an abundance of < 1 per 10,000 reads were removed [39–41]. Sample replicates were equally split between two sequence runs to prevent batch effects. The data from each run were pooled for data analysis.

## Data wrangling and formatting

The taxonomic data were converted to relative abundances and wrangled using phyloseq [42]. Relative abundances were converted to count data by multiplying by the final number of sequence reads in each sample. Reference data was uploaded in GitHub at [https://github.com/Derek-Newberger/Plant_Neighbor_Rhizosphere.git] and [https://github.com/DanielManter-USDA/DRN-2381389].

## Beta dispersion and differential abundance

Betadisper (Vegan package) was used to measure the homogeneity of multivariate dispersions [43]. Tukey Honestly Significant Difference (HSD) compared multiple beta dispersion values between plant density, plant diversity, and plant mixture treatments. Differential abundances between groups were calculated on the taxonomic species counts using the microbiomeMarker [44] package (log normalization, Benjamini-Hochberg adjustment, $p < 0.01$). The relative log expression normalization was a Log2 fold change which represents a change in bacterial taxon abundance between two samples. Groups included density comparisons (1 vs. 24 total plants and 1 vs. 48 total plants) and diversity comparisons (1 vs. 2 plant types and 1 vs. 3 plant types) within plant species (alfalfa, brassica, fescue). Additionally, similar density and diversity comparisons were made with plant species combinations. A table of significant markers was created for each comparison using the marker_table function. The relevant marker tables for density and diversity comparisons were combined, and the differentially abundant species were visualized using ggplot2 [45].

## Network analysis

Network analyses and respective statistics were conducted on taxonomic species' relative abundances using the microeco and igraph packages [46,47]. Networks for each diversity treatment (alfalfa, brassica, fescue, alfalfa-brassica, alfalfa-fescue, brassica-fescue, and alfalfa-brassica-fescue) were created with densities combined for a total of seven networks. The trans_network function was used to calculate Spearman correlations with a filter threshold of 0.001. The networks were constructed using the cal_network function with a p-value threshold of 0.01 and correlation threshold of 0.5. Network modules were partitioned using the cal_module function and the "cluster fast greedy" method. The node properties, edge properties, and adjacency matrix were obtained for each network using the get_node_table, get_edge_table, and get_adjacency_matrix functions, respectively. Networks were formatted by the rgexf package [48] and then exported to Gephi for network visualization [49]. In Gephi, networks were run using the Fruchterman Reingold algorithm with node partition colored by module, size set by relative abundance, and edges labeled by positive or negative correlations.

# Results

## The effect of intra- and inter-specific competition on the alfalfa rhizosphere

Beta dispersion values were found to be significantly different between low and high plant densities. For alfalfa, beta dispersion values decreased (from 0.46 to 0.451) when grown in monoculture and under increasing plant densities (Table 1). Bacteriome variation within the single-plant rhizosphere was inversely proportional to the proximity of neighboring alfalfa plants. Differential abundance comparisons of rhizospheres between alfalfa plants grown alone and alfalfa plants grown in monocultures of 24- and 48-plant densities showed that *Pseudarthrobacter* sp. NIBRBAC000502771, *Pseudarthrobacter phenanthrenivorans*,

**Table 1. Beta Dispersion: Average distance to median for bacteriomes.**

| A1 | A24 | A48 | Ab2 | Ab24 | Ab48 | Af2 | Af24 | Af48 | Abf3 | Abf24 | Abf48 |
|---|---|---|---|---|---|---|---|---|---|---|---|
| 0.460 | 0.458 | 0.451 | 0.500 | 0.460 | 0.423 | 0.484 | 0.467 | 0.471 | 0.493 | 0.494 | 0.488 |
| **B1** | **B24** | **B48** | **Ba2** | **Ba24** | **Ba48** | **Bf2** | **Bf24** | **Bf48** | **Baf3** | **Baf24** | **Baf48** |
| 0.464 | 0.453 | 0.483 | 0.472 | 0.424 | 0.410 | 0.524 | 0.444 | 0.425 | 0.480 | 0.497 | 0.451 |
| **F1** | **F24** | **F48** | **Fa2** | **Fa24** | **Fa48** | **Fb2** | **Fb24** | **Fb48** | **Fab3** | **Fab24** | **Fab48** |
| 0.465 | 0.475 | 0.502 | 0.501 | 0.488 | 0.476 | 0.506 | 0.483 | 0.467 | 0.481 | 0.491 | 0.473 |

Bolded capitalized letter denotes the plant species' rhizosphere (A: alfalfa, B: brassica, F: fescue). Lowercase letter denotes neighboring plant species (e.g., Ab: alfalfa rhizosphere with brassica as a plant neighbor, Abf: alfalfa rhizosphere with brassica and fescue as plant neighbors). Number denotes density (i.e., plant count per pot; low-density: 1–3 plants, medium-density: 24 plants, high-density: 48 plants). Tukey Honestly Significant Difference (HSD) showed a significant difference for the density treatment ($p = 0.013$), with high-low densities being significant (0.013), and high-medium ($p = 0.335$) and low-medium ($p = 0.09$) as non-significant. The diversity treatment was not found to be significantly different ($p = 0.072$) with no interaction between the density and diversity treatment ($p = 0.327$).

*Pseudarthrobacter oxydans*, *Neorhizobium* sp. SOG26, *Adhaeribacter swui*, *Arthrobacter* spp. UKPF54-2, and KBS0702 were consistently present in single individuals and increased in abundance with density (Fig 1). In 2-plant species mixtures, beta dispersion of the alfalfa rhizosphere decreased as total plant density increased with brassica as a neighbor (from 0.500 to 0.423) and fescue as a neighbor (from 0.484 to 0.471) (Table 1). When alfalfa was grown together with both brassica and fescue (i.e., 3-plant species mixtures), the beta dispersion of the alfalfa rhizosphere was higher compared to an alfalfa plant grown alone in low density (from 0.460 to 0.493), medium density (from 0.458 to 0.494), and high density (from 0.451 to 0.488) (Table 1).

Differential abundance comparisons of alfalfa plant rhizospheres grown alone were compared to those grown in alfalfa-brassica mixtures, alfalfa-fescue mixtures, and alfalfa-brassica-fescue mixtures across the different densities. In alfalfa rhizosphere soil grown in polyculture, eight bacterial species were significantly ($p < 0.05$) increased with a log2FC ranging from 7.59 to 20.86 (Fig 2, S2–S5 Tables). *Paucimonas lemoignei* was enriched for every treatment apart from medium-density alfalfa-brassica-fescue mixtures compared to alfalfa grown alone (Fig 2). *Adhaeribacter swui* was present in the rhizosphere of alfalfa plants grown alone and significantly enriched in the alfalfa rhizosphere in all neighbor combinations (Fig 2). *Pseudarthrobacter* sp. NIBRBAC000502771 was present in the alfalfa rhizosphere in all neighbor combinations except low-density (2- to 3- plant) alfalfa-brassica mixtures, low-density (3-plant) alfalfa-brassica-fescue mixtures, and medium-density (24) alfalfa-fescue mixtures (Fig 2). *Pseudarthrobacter phenanthrenivorans* and *Arthrobacter* sp. KBS0702 were enriched in the alfalfa rhizosphere of all neighbor combinations except low-density mixtures (Fig 2). Bacteria which were found to conditionally associate with alfalfa were *Pseudarthrobacter oxydans* (enriched in monoculture and only in plant densities of 48 plants), *Arthrobacter sp.* UKPF54-2 (enriched in monoculture and only in crop mixtures with brassica), and *Neorhizobium* sp. SOG26 (enriched in monoculture and inconsistently in plant mixtures).

Following differential abundance comparisons, a network analysis was conducted to identify microbial interactions within the alfalfa rhizosphere. *Mesobacillus subterraneus* had the highest relative abundance and was correlated with only *Mesobacillus foraminis* for almost every network. The alfalfa monoculture network had two modules tied for the most prominent module (17.31%), one of which consisted mainly of *Mesobacillus* spp. The other most prominent module (17.31%) mainly consisted of *Bacillus* spp., *Cohnella* spp., and *Paenibacillus* spp. (Fig 3a). The third most prominent module (13.46%) in the network consisted of *Arthrobacter* sp. QXT-31, *Pseudarthrobacter* sp. NIBRBAC000502771, *Arthrobacter* sp. KBS0702, *Arthrobacter* sp. UKPF54-2, *Arthrobacter* sp. PGP41, and *Pseudarthrobacter*

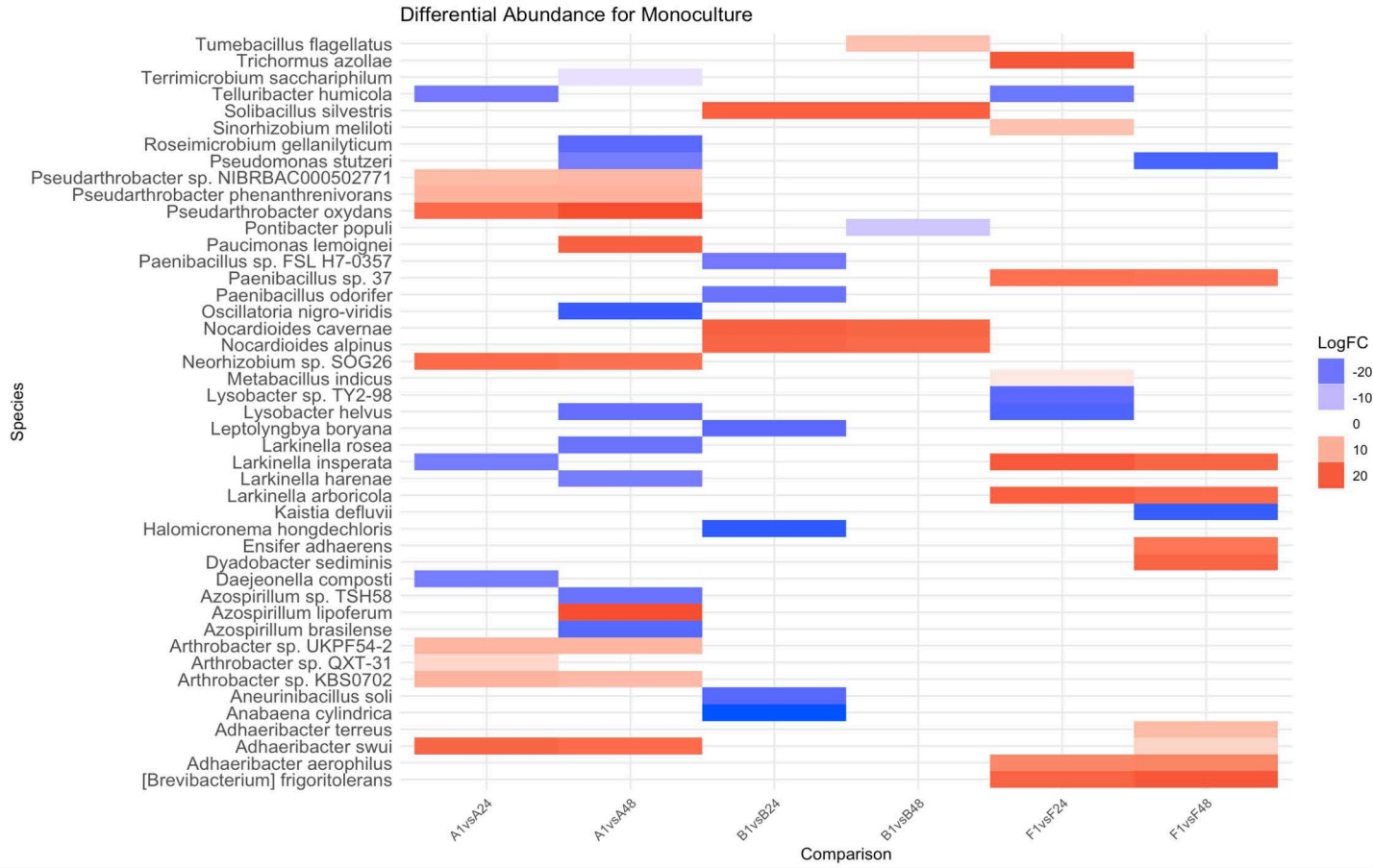

**Fig 1. Differential abundance comparisons between an individual plant (alfalfa, brassica, and fescue) rhizosphere and rhizospheres of plants grown in medium density (24 plants) and high density (48 plants) monocultures.** On the horizontal axis, the first letter denotes plant species (A: alfalfa, B: brassica, F: fescue) and number denotes the total number of plants per treatment (1 plant, 24 plants, and 48 plants). Blue denotes a negative log2 fold change ratio (decrease in abundance), while red denotes a positive log2 fold change ratio (increase in abundance).

*phenanthrenivorans*. The alfalfa-brassica plant mixture most prominent module (13.43%) primarily consisted of *Microvirga* spp. and *Flavisolibacter* spp. (Fig 3b). The alfalfa-fescue plant mixture most prominent module (16.22%), primarily consisted of *Mesobacillus* spp. and *Bacillus* spp. (Fig 3c). The alfalfa-brassica-fescue plant mixture most prominent module (14.55%) contained *Massilia* spp. (Fig 3d).

## The effect of intra- and inter-specific competition on the brassica rhizosphere

The beta dispersion of brassica increases when grown in monocultures of increasing plant densities (from 0.464 to 0.483) (Table 1). Differential abundance comparisons of the rhizosphere of a brassica plant grown alone compared to the rhizosphere of individual brassica plants grown in monoculture in higher densities showed an enrichment of *Nocardioides alpinus*, *Nocardioides cavernae*, and *Solibacillus silvestris* for the 24 and 48 density treatments (Fig 1). Differential abundance of brassica plant rhizospheres grown alone were compared to those grown in brassica-alfalfa mixtures, brassica-fescue mixtures, and brassica-alfalfa-fescue mixtures across the different densities (S6–S10 Tables)

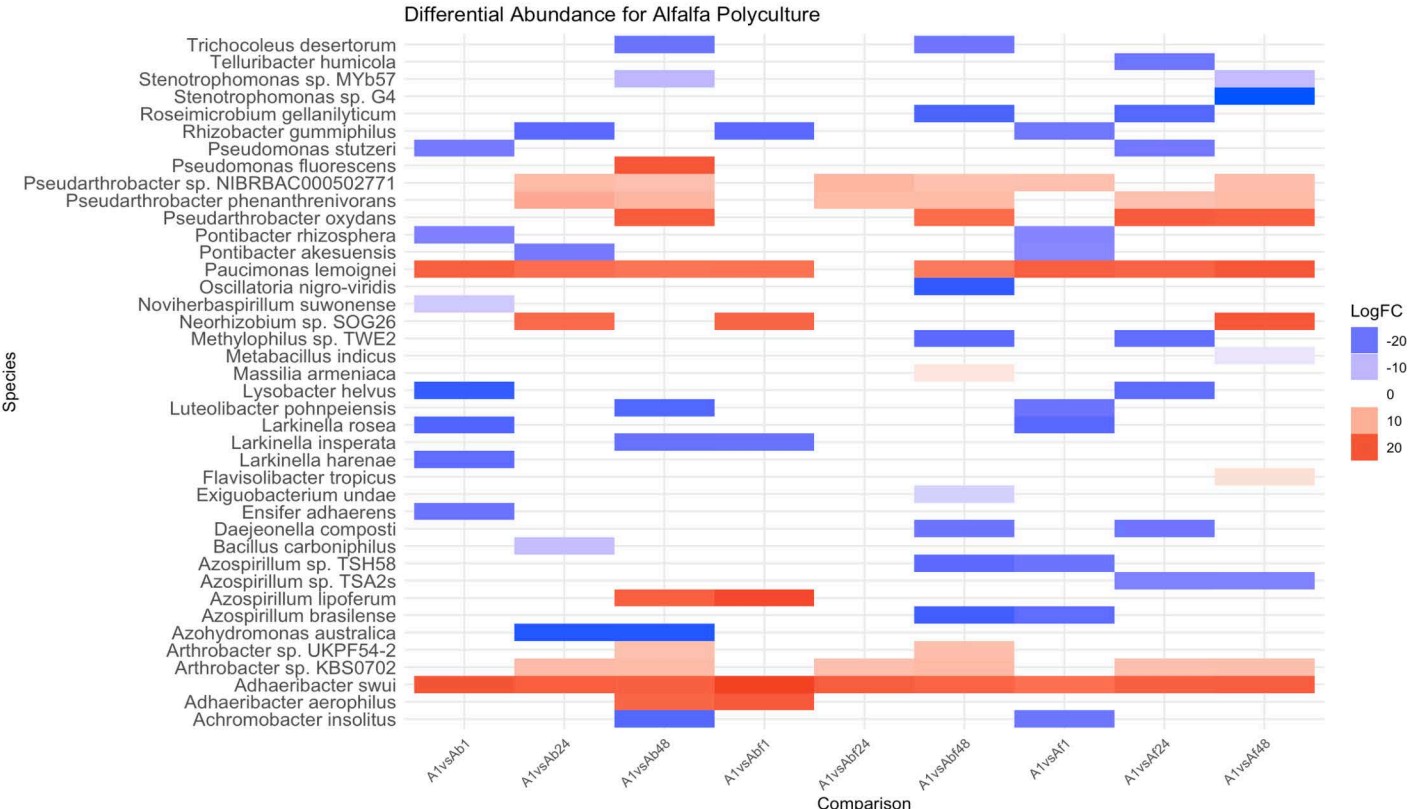

**Fig 2. Differential abundance comparisons of alfalfa rhizospheres when grown alone (1 plant) versus in medium density (24 plants) and high density (48 plants) polycultures.** On the horizontal axis, the first letter denotes plant species (A: alfalfa rhizosphere), the lowercase letter denotes neighboring plant species (Ab: alfalfa rhizosphere with neighboring brassica, Af: alfalfa rhizosphere with neighboring fescue, Abf: alfalfa rhizosphere with neighboring brassica and fescue). Number denotes the total number of plants per treatment (1 plant, 24 plants, and 48 plants). For example, Ab24 denotes alfalfa rhizosphere, with brassica as a plant neighbor, and 24 plants total with 12 being alfalfa and 12 being brassica. Blue denotes a negative log2 fold change ratio (decrease in abundance), while red denotes a positive log2 fold change ratio (increase in abundance).

In plant mixtures, the beta dispersion of brassica was significantly lower for high plant densities than low plant densities (Table 1). Differential abundance comparisons were made between the rhizosphere of brassica plants grown alone and the rhizosphere of brassica plants grown with another plant.

For brassica rhizosphere soil grown in polyculture, four bacterial species were significantly (p < 0.05) increased with a log2FC ranging from 15.39 to 20.04 (Fig 4, S7–S9 Tables). The brassica rhizosphere was strongly associated with *Nocardioides alpinus* (enriched in all treatments except for low-density brassica-fescue), *Nocardioides cavernae* (enriched in all treatments except for low-density brassica-alfalfa-fescue), and *Solibacillus silvestris* (enriched in all treatments except for low-density brassica-alfalfa-fescue and medium-density brassica-fescue) (Fig 4). *Spirosoma linguale* was enriched in low-density brassica-alfalfa mixtures, high-density brassica-alfalfa mixtures, high-density brassica-alfalfa-fescue mixtures, and low-density brassica-fescue mixtures (Fig 4).

Brassica network analysis displayed many of the same taxa as for alfalfa, but there were shifts in module connectivity and composition. In the brassica monoculture network, the most prominent module (22.41%) displayed mainly *Mesobacillus, Bacillus,* and *Paenibacillus*

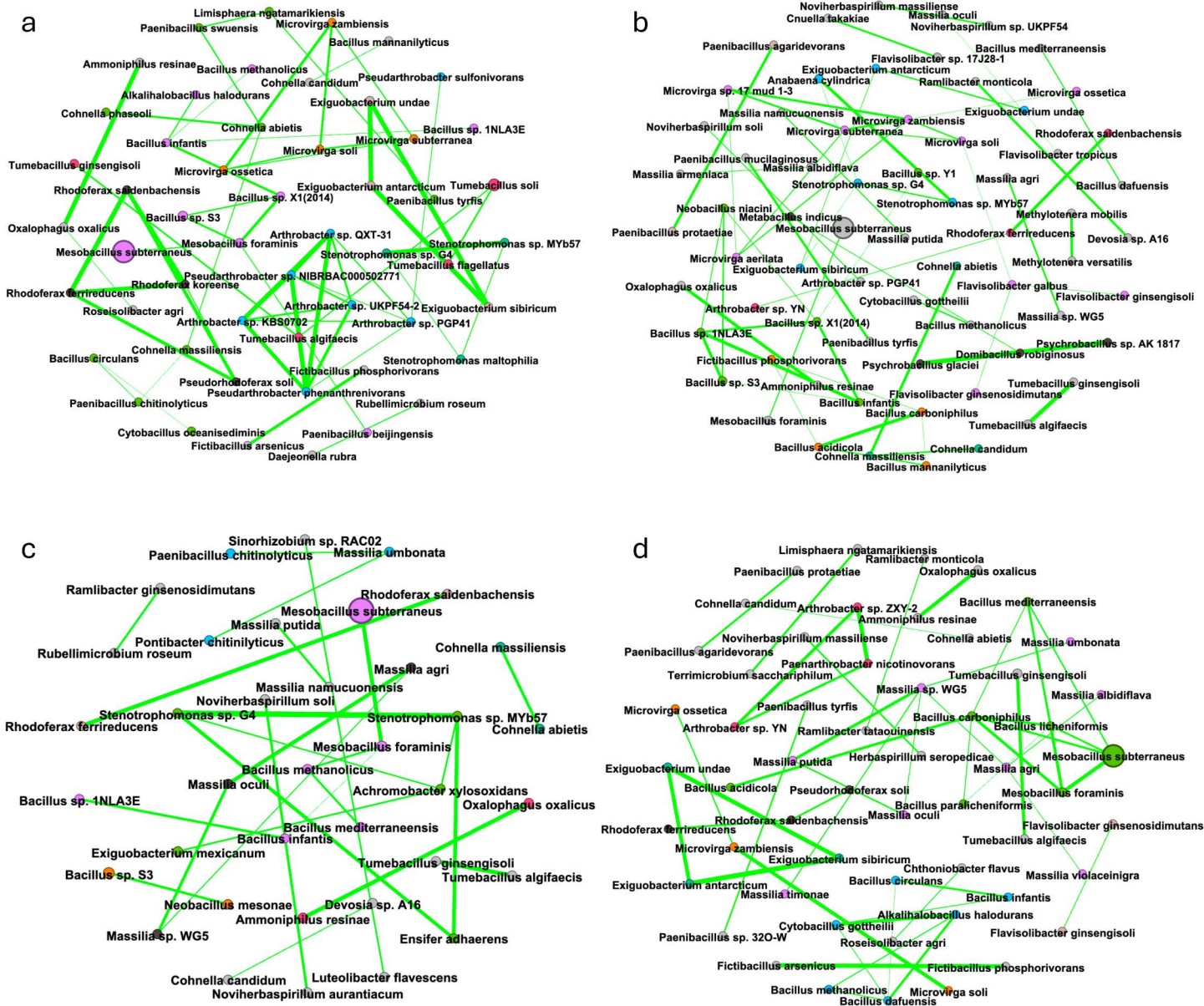

**Fig 3. Bacteriome network of alfalfa in monoculture and plant mixtures.** (a) Alfalfa monoculture (1 plant, 24 plants, 48 plants). (b) Alfalfa-brassica plant mixture (2 plants, 24 plants, 48 plants). (c) Alfalfa-fescue plant mixture (2 plants, 24 plants, 48 plants). (d) Alfalfa-brassica-fescue plant mixture (3 plants, 24 plants, 48 plants). Module color denotes module size (largest to smallest: lavender, lime green, sky blue, dark grey, orange, salmon, teal, grey); dot size denotes relative abundance; green edge denotes a positive correlation; red edge would denote a negative correlation if present.

spp. (Fig 5a). The brassica-alfalfa mixture most prominent module (14.89%) displayed mainly *Bacillus* spp. (Fig 5b). The brassica-fescue mixture showed the most prominent module (23.33%) and mainly displayed *Mesobacillus, Cytobacillus, Bacillus,* and *Neobacillus* spp. (Fig 5c). The brassica-alfalfa-fescue plant mixture network's most prominent module (18.87%) mainly displayed *Mesobacillus* and *Bacillus* spp. (Fig 5d). It is curious to note that the only negative correlation identified in this study is between *Bacillus* sp. S3 and *Flavisolibacter ginsengiterrae* as depicted by the red line in Fig 5a.

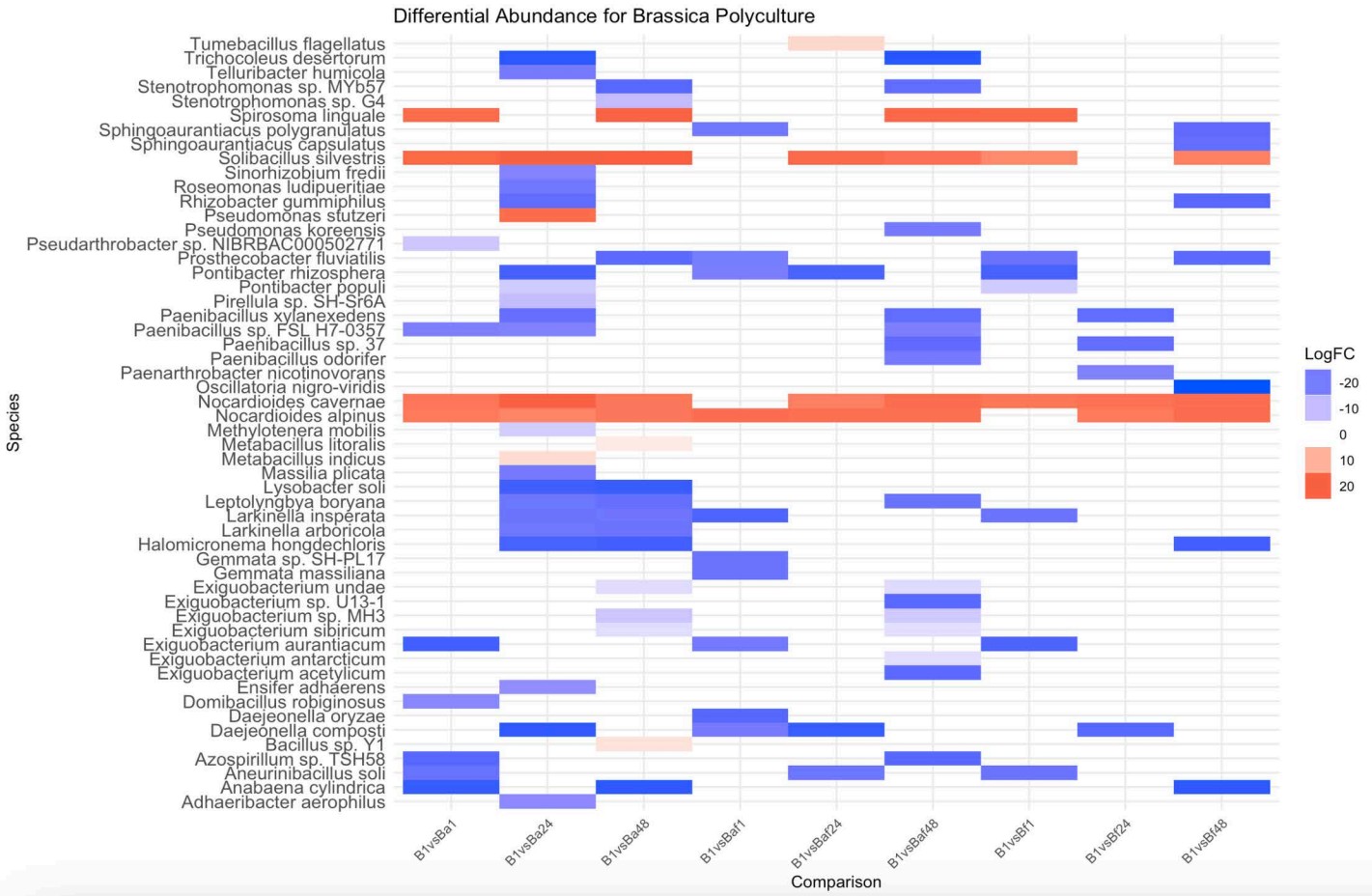

**Fig 4. Differential abundance comparison of the rhizosphere of a brassica plant grown alone and brassica's rhizosphere grown in medium density (24 plants) and high density (48 plants) polycultures.** On the horizontal axis, the first letter denotes plant species (B: brassica rhizosphere), the lowercase letter denotes neighboring plant species (Ba: brassica rhizosphere with neighboring alfalfa, Bf: brassica rhizosphere with neighboring fescue, Baf: brassica rhizosphere with neighboring alfalfa and fescue). Number denotes the total number of plants per treatment (1 plant, 24 plants, and 48 plants). For example, Ba24 denotes brassica rhizosphere, with alfalfa as a plant neighbor, and 24 plants total with 12 being brassica and 12 being alfalfa. Blue denotes a negative log2 fold change ratio (decrease in abundance), while red denotes a positive log2 fold change ratio (increase in abundance).

## The effect of intra- and inter-specific competition on the fescue rhizosphere

The beta dispersion of fescue increased when grown in monoculture of increasing plant densities (from 0.465 to 0.502) (Table 1). Differential abundance comparisons of fescue plant rhizospheres grown alone were compared to those grown in fescue-alfalfa mixtures, fescue-brassica mixtures, and fescue-alfalfa-brassica mixtures across the different densities (S11–S13 Tables). Differential abundance comparisons between the rhizosphere of a fescue plant grown alone and the rhizosphere of individual fescue plants grown in monoculture densities of 24 and 48 plants demonstrated an enrichment of *Adhaeribacter aerophilus*, *[Brevibacterium] frigoritolerans*, *Larkinella arboricola*, *Larkinella insperata*, and *Paenibacillus* sp. 37 for the higher density treatments (Fig 1). *Ensifer adhaerens* and *Dyadobacter sediminis* were enriched in medium and high fescue densities compared to low plant densities (Fig 1).

Within the fescue rhizosphere, strong correlations were seen for *Adhaeribacter aerophilus* (enriched for all treatments except low-density fescue-alfalfa), *Larkinella arboricola*

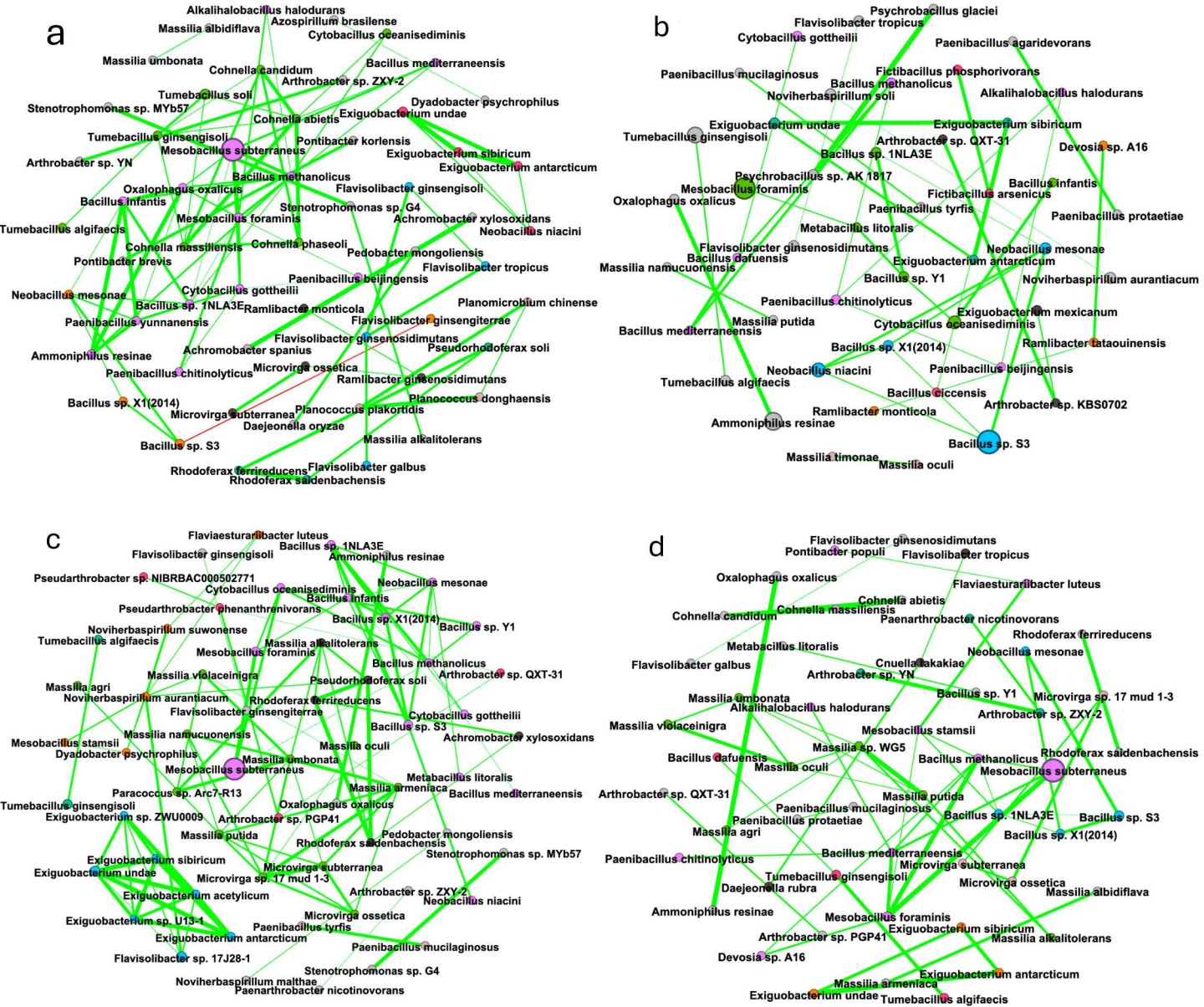

**Fig 5. Bacteriome network of brassica in monoculture and plant mixtures.** (a) Brassica monoculture (1 plant, 24 plants, 48 plants). (b) Brassica-alfalfa plant mixture (2 plants, 24 plants, 48 plants). (c) Brassica-fescue plant mixture (2 plants, 24 plants, 48 plants). (d) Brassica-alfalfa-fescue plant mixture (3 plants, 24 plants, 48 plants). Module color denotes module size (largest to smallest: lavender, lime green, sky blue, dark grey, orange, salmon, teal, grey), size of the dot denotes relative abundance; green edge denotes a positive correlation, and red edge denotes a negative correlation.

(enriched for all treatments except low-density fescue-alfalfa-brassica), *Larkinella insperata* (enriched for all treatments except low-density fescue-alfalfa-brassica), *Dyadobacter sediminis* (except for medium-density brassica), *Paenibacillus* sp. 37 (enriched for all treatments except low-density fescue-alfalfa and high-density fescue-alfalfa-brassica), and *Ensifer adhaerens* (enriched for all treatments except for medium-density brassica, low-density fescue-alfalfa-brassica, and medium-density fescue-brassica).

In plant paired mixtures, the beta dispersion of fescue decreased as plant density increased when grown with either alfalfa (from 0.501 to 0.476) or brassica (from 0.506 to 0.467)

(Table 1). In mixtures with all three plants, the beta dispersion of microcosms of three plants (low-density) (Abf3: 0.493, Baf3: 0.480, Fab24: 0.481) was higher than microcosms of their respective plant diversity at 48 plants (Abf48: 0.488, Baf48: 0.451, Fab48: 0.473) (Table 1). For medium (24 plants) densities, the beta dispersion value was higher for plant mixtures of three plants than alfalfa-brassica plant mixtures (Table 1). For high fescue densities (48 plants), the beta dispersion value for mixtures of three plants (Fab48: 0.473) was higher than fescue-brassica mixtures (Fb48: 0.467) (Table 1). For fescue rhizosphere soil grown in polyculture, seven bacterial species were significantly ($p < 0.05$) increased with a log2FC ranging from 14.96 to 20.90 (Fig 6, S11–S13 Tables). *Larkinella arboricola* and *Larkinella insperata* were enriched in rhizospheres of all crop mixtures except for low-density (three plants) fescue-alfalfa-brassica mixtures compared to the rhizosphere of a fescue plant grown alone (Fig 6). Differential abundance comparisons showed *Adhaeribacter aerophilus* was enriched in the rhizosphere of individual fescue plants grown in 24 fescue-alfalfa mixtures, 48 fescue-alfalfa mixtures, and all fescue-brassica/fescue-alfalfa-brassica mixtures compared to the rhizosphere of an individual fescue plant (Fig 6). Differential abundance comparisons showed *[Brevibacterium] frigoritolerans* was enriched in the rhizosphere of individual fescue plants grown in all medium-density plant mixtures (fescue-alfalfa, fescue-brassica, fescue-alfalfa-brassica) and in high-density fescue-brassica mixtures compared to the rhizosphere of an individual fescue plant (Fig 6). *Paenibacillus* sp. 37 (Fig 6) was enriched in the rhizosphere of fescue plants in all

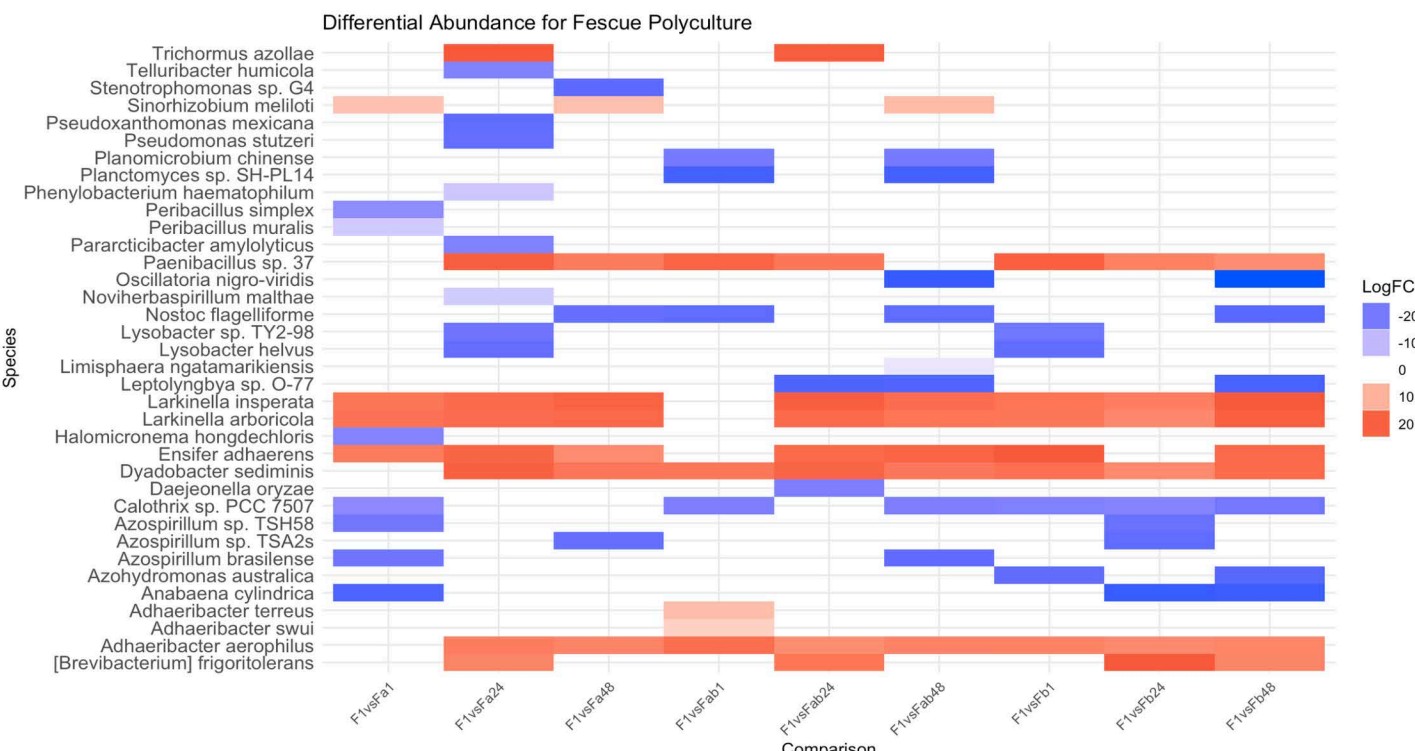

**Fig 6. Differential abundance comparison of the rhizosphere of a fescue plant grown alone and fescue's rhizosphere grown in medium density (24 plants) and high density (48 plants) polycultures.** On the horizontal axis, the first letter denotes plant species (F: fescue rhizosphere), the lowercase letter denotes neighboring plant species (Fa: fescue rhizosphere with neighboring alfalfa, Fb: fescue rhizosphere with neighboring brassica, Fab: fescue rhizosphere with neighboring alfalfa and brassica). Number denotes the total number of plants per treatment (1 plant, 24 plants, and 48 plants). For example, Fa24 denotes fescue rhizosphere, with alfalfa as a plant neighbor, and 24 plants total with 12 being fescue and 12 being alfalfa. Blue denotes a negative log2 fold change ratio (decrease in abundance), while red denotes a positive log2 fold change ratio (increase in abundance).

treatments except in low-density fescue-alfalfa mixtures (two plants) and high-density fescue-alfalfa-brassica mixtures (48 plants). Differential abundance comparisons showed *Ensifer adhaerens* was enriched in the rhizosphere of all fescue crop mixtures except for low densities of fescue-alfalfa-brassica mixtures and medium densities of fescue-brassica mixtures compared to the rhizosphere of an individual fescue plant (Fig 6). Differential abundance comparisons showed *Dyadobacter sediminis* was enriched in the rhizosphere of all fescue crop mixtures except for low densities of fescue-alfalfa mixtures compared to the rhizosphere of an individual fescue plant (Fig 6).

Network analysis was also run for the fescue rhizosphere for the different plant combinations. Fescue monoculture's most prominent module (12.5%) only consisted of *Microvirga* spp. (Fig 7a). For the fescue-alfalfa mixture network, the most prominent module (19.15%) mainly consisted of *Bacillus* and *Mesobacillus* spp. (Fig 7b). In the fescue-brassica mixture, the most prominent module (20.73%) mostly consisted of *Bacillus*, *Cytobacillus*, and *Mesobacillus* spp. (Fig 7c). The fescue-alfalfa-brassica plant mixture network's most prominent module (20%) consisted mainly of *Bacillus* spp.

## Discussion

It is well established that an individual plant's growth is affected by its plant neighbors through interspecific and intraspecific competition. Plant-plant competition can be manifested as growing longer roots to increase access to nutrients or growing taller to access higher quality sunlight [50]. This dominance should also be measured by how impactful the plant's root exudates are on microbial recruitment in the rhizosphere. Since the bacteriome of the rhizosphere is critical for the plant's development and stress tolerance [51], it is important to acknowledge the influence that a plant neighbor has on rhizosphere colonization. In the present study, the alfalfa monoculture rhizospheres showed a decrease in variability of microbial diversity as plant density increased. In contrast, brassica and fescue monoculture rhizospheres showed an increase in microbial diversity variability with increasing plant density. It is possible that alfalfa's intraspecific allelopathic ability could have had an influence in reducing the variability of microbial diversity of the rhizosphere. Alfalfa secretes autotoxic chemicals which reduce the establishment of new alfalfa seedlings [52,53]. Although these plant-derived chemicals do not appear to negatively impact adult alfalfa stands [53], these chemicals could negatively impact the soil health in the long term since they may increase pathogenic fungi and decrease beneficial microorganisms [54]. For brassica, cover crop *Brassica juncea* was successfully used as a biocontrol since it decreased *Escherichia coli* populations to non-detectable levels in a greenhouse study [55]. *B. juncea*, when used as green manure for cucumbers, altered the mycobiome composition of the rhizosphere without changing the alpha diversity [56]. This finding is reflective in the present study as well since brassica did not decrease the variability of the rhizosphere's bacteriome but caused a shift in its composition.

Microbial beta dispersion analysis showed that plant density was significantly different (p = 0.013) between high and low plant densities. As plant density increased in plant mixtures, the microbial variability of the plant's rhizosphere decreased within each plant diversity treatment. This supports the notion that increasing the number of plant neighbors reduces the rhizobacterial diversity for each plant. Although non-significant (0.072), plant diversity increases the diversity of bacteria recruited to the rhizosphere. Overall, plant density was shown to be a more important driver for microbial recruitment than plant diversity as shown by the beta dispersion.

Differential abundance analysis unveiled bacterial taxa that strongly associate with the rhizospheres of alfalfa, brassica, and fescue. However, bacterial taxa which were no longer enriched in the rhizosphere if density or diversity changed are considered to express

conditional associations with their specific plant host. Each of these plant species tested has been known to affect the soil microbiome. Allelochemicals from alfalfa plants increase pathogenic fungi and decrease beneficial microorganisms [54], and this microbial shift could negatively impact any plant neighbor. *Brassica juncea* is known to have antimicrobial and insecticidal properties [57]. Since antimicrobials of brassica impede certain microorganisms [30], it can potentially influence the recruitment of neighboring plants. It has been shown that *Festuca* sp. is allelopathic and can outcompete sweetgum by reducing sweetgum biomass

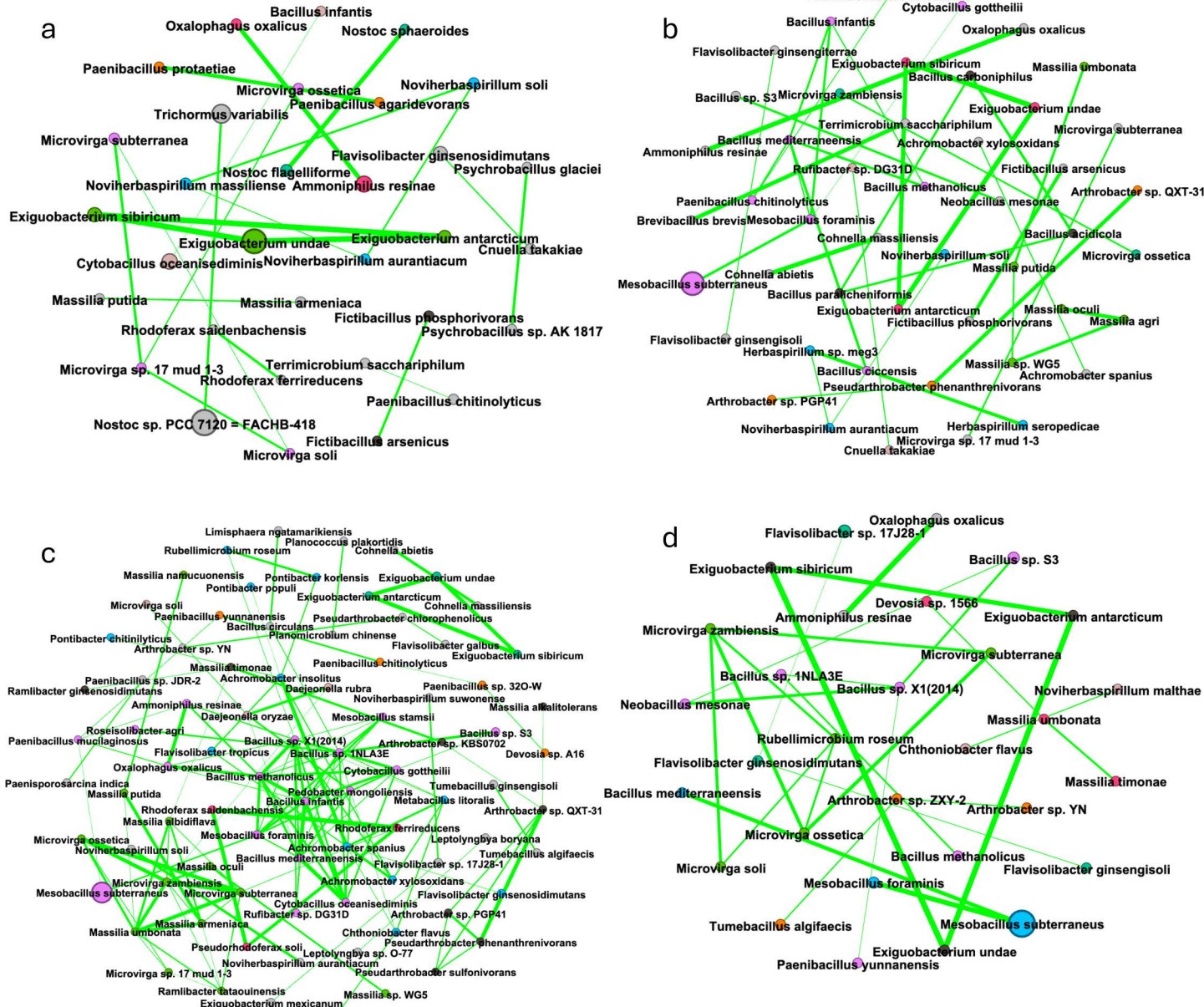

**Fig 7. Bacteriome network of brassica in monoculture and plant mixtures.** (a) Fescue monoculture (1 plant, 24 plants, 48 plants). (b) Fescue-alfalfa plant mixture (2 plants, 24 plants, 48 plants). (c) Fescue-brassica plant mixture (2 plants, 24 plants, 48 plants). (d) Fescue-alfalfa-brassica plant mixture (3 plants, 24 plants, 48 plants). Module color denotes module size (largest to smallest: lavender, lime green, sky blue, dark grey, orange, salmon, teal, grey), size of the dot denotes relative abundance; green edge denotes a positive correlation, and a red edge would denote a negative correlation if present.

[58]. In addition, *Festuca* sp. has expressed competitive ability against red clover (*Trifolium pratense L.*) which is dependent on the colonization of *Neotyphodium* endophytes [59].

*Pseudarthrobacter* sp. NIBRBAC000502771, *Pseudarthrobacter phenanthrenivornans*, and *Arthrobacter* sp. KBS0702 are present only in high alfalfa plant densities, and they remain enriched despite whether the plant neighbor is a different species. For *Pseudarthrobacter oxydans*, density was more influential than diversity. Enrichment of *Arthrobacter sp.* UKPF54-2 and *Neorhizobium* sp. SOG26 could have been diversity dependent with enrichment of *A.* UKPF54-2 either negatively influenced by fescue or positively influenced by brassica. *Neorhizobium* sp. SOG26 was negatively influenced by brassica and fescue. Past studies have shed some light onto the possible functions of those strongly correlated rhizobacteria of alfalfa (see S15 for a list of functions) [60–65]. Of the bacterial taxa that were found to be differentially abundant between alfalfa grown alone and with a plant neighbor, three are capable of nitrogen fixation and three produce some form of phytohormone. Although *Pseudoarthrobacter oxydans* can perform nitrogen fixation and produce phytohormones, it can perform a wider range of functions as shown in the studies. There may additionally unknown functions that explain its association with alfalfa.

The brassica rhizosphere was strongly associated with *Nocardioides alpinus*, *Nocardioides cavernae*, and *Solibacillus silvestris*. *Spirosoma linguale* was conditionally associated with brassica and was possibly diversity dependent since it was enriched in every crop mixture treatment apart from low-density brassica-alfalfa-fescue and medium densities of brassica-fescue. Previous studies have reviewed possible functions of these strongly associated brassica rhizobacteria. *Nocardioides alpinus* strains have been found to reduce nitrate and produce ammonia and indole-3-acetic acid [66,67]. The purpose of *Nocardioides cavernae* recruitment is not indicated by the literature, since this taxon does not have nitrogen, phosphorus, or indole related activities [68]. *Solibacillus silvestris* may show N-Acyl homoserine lactone degrading activity which has quorum-quenching and biocontrol activities [69]. In addition, *Solibacillus silvestris* may produce phytohormones (indole-3-acetic acid, cytokinin, and gibberellin), fix nitrogen, and be resistant to cadmium [70].

Within the fescue rhizosphere, [*Brevibacterium*] *frigoritolerans* and *Trichormus azollae* may have been outcompeted by microbes from other plant species, especially when planted in higher densities. Rhizobacterial enrichment even of the strongly correlated taxa needed more than one fescue plant to be present. [*Brevibacterium*] *frigoritolerans* (enriched in monoculture, all medium-density polyculture, and high-density fescue-brassica mixtures), *Trichormus azollae* (enriched in all medium densities except for fescue-brassica mixtures), and *Sinorhizobium meliloti* (enriched inconsistently, but only for mixtures including alfalfa) all showed fewer correlations and thus possibly a conditional association with the fescue rhizosphere. Plausible functions of the fescue rhizosphere bacterial taxa were searched for in the literature (see S14 Table for a list of functions) [71–80]. Of interest, *Ensifer adhaerens* (found in fescue) is a nitrogen fixer and may produce indole-3-acetic acid, exopolysaccharides, ammonia, siderophores, salicylic acid (for abiotic stress), and even promote seed germination for soybean [79]. Another legume associated bacteria, *Sinorhizobium meliloti,* which is a nitrogen fixing symbiont of alfalfa, was found in the rhizosphere of the fescue sp. [80]. Past studies have shown *Ensifer adhaerens* and *Sinorhizobium meliloti* as part of the legume rhizosphere, but not the fescue rhizosphere, which supports the sharing of microbes between alfalfa and fescue. *E. adhaerens* was not enriched for only mixtures which included brassica, meanwhile *S. meliloti* was only enriched in mixtures which included alfalfa.

Network analysis did not show more edges and nodes as plant diversity increased, and taxa with high abundances were not necessarily hub species (S15 and S16 Tables) [81]. For alfalfa only, taxa in networks overlapped with taxa that were differentially abundant. This supports

that these bacteria not only have a strong association with alfalfa, but between each other as well. Rhizobacterial networks which had the highest number of modules (Fb: 82, Ab: 71, Bf: 60) and edges (Fb: 136, Bf:99) were from plant mixtures which included brassica. Modularity (microbial independent structural units) in crop mixtures was highest for alfalfa in alfalfa-brassica (0.898), for brassica in brassica-alfalfa (0.861), and for fescue in fescue-alfalfa (0.898). Increasing agricultural intensification has shown to decrease microbial network complexity [82,83], which contrasts our study which showed that higher plant diversity increased bacterial modules, and therefore complexity. *Mesobacillus subterraneus* had a high relative abundance but was rarely connected to more than 1-2 taxa (Figs 3a, 3b, 3c, 5c, 7b, 7c, and 7d). This finding implies that relative abundance alone does not serve as a good predictive measurement for a keystone bacterial species and therefore is not expected to drive microbial shifts in the rhizosphere. The bacterial family Bacillaceae dominated the networks. This dominance could possibly be explained by how many members of Bacillaceae are thermophiles [84,85]. Accordingly, the method by which the microbial complexity was reduced in this study was an autoclave which may have inadvertently selected for thermophilic *Bacillaceae*. Nevertheless, modules mainly composed of *Microvirga* sp. and *Massilia* sp. were still able to thrive and become the predominant modules over members of the Bacillaceae family, most likely due to plant host interaction. Plants recruit a specific set of microorganisms as shown through differential abundance comparisons, however, by increasing plant diversity these recruited bacteria no longer played a major role in the rhizobacterial network. Alfalfa's third most prominent module in monoculture contained some of the same taxa that were shown to be strongly associated with the rhizosphere in our differential abundance analysis. These taxa included *Pseudarthrobacter* sp. NIBRBAC000502771, *Arthrobacter* sp. KBS0702, *Arthrobacter* sp. UKPF54-2, and *Pseudarthrobacter phenanthrenivorans*. Increasing interspecific competition aided the recruitment of these bacteria. However, the neighboring bacteriome networks of different plant neighbors did not appear to have much interaction.

## Conclusion

The present shared rhizosphere microcosm study which used *Medicago sativa*, *Brassica* sp., and *Fescue* sp. found that rhizobacterial beta diversity typically decreased as plant density increased. In addition, plant density was a stronger driver for beta diversity than plant diversity. Independent of plant neighbor identity or density, each target plant species consistently recruited a low number of specific rhizobacteria. However, some bacterial taxa were shown to be enriched within only high plant density treatments. Bacterial recruitment by plants is selective, indicating that these taxa may play a role to alleviate plant-plant competition. Furthermore, there was evidence of bacterial sharing of nitrogen fixers from alfalfa to fescue. From an agriculturalist perspective, increasing the number of the same plant species could help each plant recruit their preferred rhizobacteria faster, as is done in monoculture. In addition, planting a different plant species as a neighbor, as done in polyculture, does not seem to greatly hinder the bacterial recruitment of either plant species. This supports that interspecific competition through polycultures and intercropping is not detrimental to bacterial recruitment and could support co-existence of plant species.

## Supporting information

**S1 Table. Experimental setup of the increasing plant densities and diversities.** This table shows the plant count for each of the 21 treatments with the black boarder representing each pot. From these 21 treatments, there were 36 different rhizosphere samples as noted by the labels in bold.
(PDF)

**S1 Protocol. Library preparation and sequencing for MinION – Manter Lab Protocol.** (PDF)

**S2 Table. Differential abundance comparison of alfalfa when grown alone (1 plant) and alfalfa plant densities.** Enriched column shows which treatment the bacterial taxa is enriched (A1: single alfalfa plant, A24: 24 alfalfa plants, A48: 48 alfalfa plants). Bacterial taxa which were enriched in only one treatment of increasing plant density is highlighted in orange. Bacterial taxa which were enriched all density treatment is highlighted in sky blue. (PDF)

**S3 Table. Differential abundance comparison of alfalfa when grown alone (1 plant) and alfalfa-brassica mixtures.** Enriched column shows which treatment the bacterial taxa is enriched (A1: single alfalfa plant, Ab2: single alfalfa and brassica plant, Ab24: 12 alfalfa and brassica plants, Ab48: 24 alfalfa and brassica plants). Bacterial taxa which were enriched when alfalfa was grown alone as compared to multiple density treatments. Bacterial taxa which were enriched in only one treatment of increasing plant density is highlighted in orange. Bacterial taxa which were enriched in more than one diversity treatment is highlighted in light sky blue. Bacterial taxa which were enriched all density treatment is highlighted in sky blue. (PDF)

**S4 Table. Differential abundance comparison of alfalfa when grown alone (1 plant) and alfalfa-fescue mixtures.** Enriched column shows which treatment the bacterial taxa is enriched (A1: single alfalfa plant, Af2: single alfalfa and fescue plant, Af24: 12 alfalfa and fescue plants, Af48: 24 alfalfa and fescue plants). Bacterial taxa which were enriched when alfalfa was grown alone as compared to multiple density treatments. Bacterial taxa which were enriched in only one treatment of increasing plant density is highlighted in orange. Bacterial taxa which were enriched in more than one diversity treatment is highlighted in light sky blue. Bacterial taxa which were enriched all density treatment is highlighted in sky blue. (PDF)

**S5 Table. Differential abundance comparison of alfalfa when grown alone (1 plant) and alfalfa-brassica-fescue mixtures.** Enriched column shows which treatment the bacterial taxa is enriched (A1: single alfalfa plant, Abf2: single alfalfa, brassica, and fescue plant, Abf24: 8 alfalfa, brassica, and fescue plants, Af48: 16 alfalfa, brassica, and fescue plants). Bacterial taxa which were enriched when alfalfa was grown alone as compared to multiple density treatments. Bacterial taxa which were enriched in only one treatment of increasing plant density is highlighted in orange. Bacterial taxa which were enriched in more than one diversity treatment is highlighted in light sky blue. Bacterial taxa which were enriched all density treatment is highlighted in sky blue. (PDF)

**S6 Table. Differential Abundance Comparison of brassica when grown alone (1 plant) and brassica plant densities.** Enriched column shows which treatment the bacterial taxa is enriched (B1: single brassica plant, Ba2: single brassica and alfalfa plants, B24: 24 brassica plants, B48: 48 brassica plants). Bacterial taxa which were enriched when brassica was grown alone as compared to multiple density treatments. Bacterial taxa which were enriched in only one treatment of increasing plant density is highlighted in orange. Bacterial taxa which were enriched in more than one diversity treatment is highlighted in light sky blue. Bacterial taxa which were enriched all density treatment is highlighted in sky blue. (PDF)

**S7 Table. Differential abundance comparison of brassica when grown alone (1 plant) and brassica-alfalfa mixtures.** Enriched column shows which treatment the bacterial taxa is enriched (B1: single brassica plant, Ba24: 12 brassica and alfalfa plants, Ba48: 24 brassica and alfalfa plants). Bacterial taxa which were enriched when brassica was grown alone as compared to multiple density treatments. Bacterial taxa which were enriched in only one treatment of increasing plant density is highlighted in orange. Bacterial taxa which were enriched in more than one diversity treatment is highlighted in light sky blue. Bacterial taxa which were enriched all density treatment is highlighted in sky blue.
(PDF)

**S8 Table. Differential abundance comparison of brassica when grown alone (1 plant) and brassica-fescue mixtures.** Enriched column shows which treatment the bacterial taxa is enriched (B1: single brassica plant, Bf1: single brassica and fescue plants, Bf24: 12 brassica and fescue plants, Bf48: 24 brassica and fescue plants). Bacterial taxa which were enriched when brassica was grown alone as compared to multiple density treatments. Bacterial taxa which were enriched in only one treatment of increasing plant density is highlighted in orange. Bacterial taxa which were enriched in more than one diversity treatment is highlighted in light sky blue. Bacterial taxa which were enriched all density treatment is highlighted in sky blue.
(PDF)

**S9 Table. Differential abundance comparison of brassica when grown alone (1 plant) and brassica-alfalfa-fescue mixtures.** Enriched column shows which treatment the bacterial taxa is enriched (B1: single brassica plant, Baf1: single brassica, alfalfa, and fescue plants, Baf24: 8 brassica, alfalfa, and fescue plants, Baf48: 16 brassica, alfalfa, and fescue plants). Bacterial taxa which were enriched when brassica was grown alone as compared to multiple density treatments. Bacterial taxa which were enriched in only one treatment of increasing plant density is highlighted in orange. Bacterial taxa which were enriched in more than one diversity treatment is highlighted in light sky blue. Bacterial taxa which were enriched all density treatment is highlighted in sky blue.
(PDF)

**S10 Table. Differential abundance comparison of fescue when grown alone (1 plant) and fescue plant densities.** Enriched column shows which treatment the bacterial taxa is enriched (F1: single fescue plant, F24: 24 fescue plants, F48: 48 fescue plants). Bacterial taxa which were enriched when fescue was grown alone as compared to multiple density treatments. Bacterial taxa which were enriched in only one treatment of increasing plant density is highlighted in orange. Bacterial taxa which were enriched in more than one diversity treatment is highlighted in light sky blue. Bacterial taxa which were enriched all density treatment is highlighted in sky blue.
(PDF)

**S11 Table. Differential abundance comparison of fescue when grown alone (1 plant) and fescue-alfalfa mixtures.** Enriched column shows which treatment the bacterial taxa is enriched (F1: single fescue plant, Fa1: single fescue and alfalfa plants, Fa24: 12 fescue and alfalfa plants, Fa48: 24 fescue and alfalfa plants). Bacterial taxa which were enriched when fescue was grown alone as compared to multiple density treatments. Bacterial taxa which were enriched in only one treatment of increasing plant density is highlighted in orange. Bacterial taxa which were enriched in more than one diversity treatment is highlighted in light sky blue. Bacterial taxa which were enriched all density treatment is highlighted in sky blue.
(PDF)

**S12 Table. Differential abundance comparison of fescue when grown alone (1 plant) and fescue-brassica mixtures.** Enriched column shows which treatment the bacterial taxa is enriched (F1: single fescue plant, Fb1: single fescue and brassica plants, Fb24: 12 fescue and brassica plants, Fb48: 24 fescue and brassica plants). Bacterial taxa which were enriched when fescue was grown alone as compared to multiple density treatments. Bacterial taxa which were enriched in only one treatment of increasing plant density is highlighted in orange. Bacterial taxa which were enriched in more than one diversity treatment is highlighted in light sky blue. Bacterial taxa which were enriched all density treatment is highlighted in sky blue.
(PDF)

**S13 Table. Differential abundance comparison of fescue when grown alone (1 plant) and fescue-alfalfa-brassica mixtures.** Enriched column shows which treatment the bacterial taxa is enriched (F1: single fescue plant, Fab1: single fescue, alfalfa, and brassica plants, Fab24: 8 fescue, alfalfa, and brassica plants, Fab48: 16 fescue, alfalfa, and brassica plants). Bacterial taxa which were enriched when fescue was grown alone as compared to multiple density treatments. Bacterial taxa which were enriched in only one treatment of increasing plant density is highlighted in orange. Bacterial taxa which were enriched in more than one diversity treatment is highlighted in light sky blue. Bacterial taxa which were enriched all density treatment is highlighted in sky blue.
(PDF)

**S14 Table. Bacterial taxa characterization for alfalfa and brassica related microbes.**
(PDF)

**S15 Table. 1st, 2nd, and 3rd largest module by plant network.**
(PDF)

**S16 Table. Bacteriome Network Statistics.** First capitalized letter denotes plant species rhizosphere; lower case letter denotes neighboring plant species.
(PDF)

## Acknowledgments

This original research article's contents are solely the responsibility of the authors and do not necessarily represent the official views of the USDA. Big thanks to Mary Dixon of the Vivanco lab, for providing the code for an example microbiome network and support on statistics. Special thanks to Timothy Creed of the USDA Soil Management and Sugar Beet Research Unit, for DNA extraction and sequencing training. Profound thanks to Madelene Shehan for providing grammar, syntax, and sentence structure edits to the manuscript.

## Author contributions

**Conceptualization:** Jorge M. Vivanco, Derek R. Newberger.

**Data curation:** Derek R. Newberger.

**Formal analysis:** Derek R. Newberger, Heather L. Deel, Daniel K. Manter.

**Funding acquisition:** Jorge M. Vivanco, Daniel K. Manter.

**Investigation:** Jorge M. Vivanco, Derek R. Newberger.

**Methodology:** Derek R. Newberger.

**Project administration:** Jorge M. Vivanco, Derek R. Newberger.

**Resources:** Jorge M. Vivanco, Daniel K. Manter.

**Software:** Daniel K. Manter.

**Supervision:** Jorge M. Vivanco.

**Validation:** Jorge M. Vivanco, Derek R. Newberger, Heather L. Deel, Daniel K. Manter.

**Visualization:** Derek R. Newberger, Heather L. Deel.

**Writing – original draft:** Jorge M. Vivanco, Derek R. Newberger.

**Writing – review & editing:** Jorge M. Vivanco, Derek R. Newberger, Heather L. Deel, Daniel K. Manter.

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
