## [Decision Letter · Decision Letter 0]

15 Nov 2024

PONE-D-24-42703Effect of intra- and inter-specific plant interactions on the rhizosphere microbiome of a single target plant at different densitiesPLOS ONE

Dear Dr. Vivanco,

Thank you for submitting your manuscript to PLOS ONE. After careful consideration, we feel that it has merit but does not fully meet PLOS ONE’s publication criteria as it currently stands. Therefore, we invite you to submit a revised version of the manuscript that addresses the points raised during the review process.

We look forward to receiving your revised manuscript.

Kind regards,

Eugenio Llorens

Academic Editor

PLOS ONE

**Journal Requirements:**

2. As noted, this manuscript relates to your previous publication in Scientific Reports. In your manuscript, please ensure you cite, discuss, and acknowledge overlap with the related work, and provide adequate justification (in the Introduction) for the new submission in light of the related published work.

This original research article was financially supported by National Institute of Food and Agriculture (NIFA)/ United States Department of Agriculture (USDA) through a Western Sustainable Agriculture Research and Education (SARE) project #SW20-910. Funding was awarded to J.M. and D.M., and the sponsors did not play any role in the study design, data collection and analysis, decision to publish, or preparation of the manuscript.

https://www.nifa.usda.gov/

https://western.sare.org/

5. Please upload a new copy of Figures 3, 5 and 7 as the detail is not clear. Please follow the link for more information: ">https://blogs.plos.org/plos/2019/06/looking-good-tips-for-creating-your-plos-figures-graphics/"
">https://blogs.plos.org/plos/2019/06/looking-good-tips-for-creating-your-plos-figures-graphics/"

Reviewers' comments:

Reviewer's Responses to Questions

**Comments to the Author**

1. Is the manuscript technically sound, and do the data support the conclusions?

Reviewer #1: Yes

Reviewer #2: Yes

2. Has the statistical analysis been performed appropriately and rigorously? 

Reviewer #1: Yes

Reviewer #2: Yes

3. Have the authors made all data underlying the findings in their manuscript fully available?

Reviewer #1: Yes

Reviewer #2: Yes

4. Is the manuscript presented in an intelligible fashion and written in standard English?

Reviewer #1: Yes

Reviewer #2: Yes

5. Review Comments to the Author

**Reviewer #1: ** This is an interesting study where the authors examine, through a pot microcosm experiment, the rhizosphere microbiome of Alfalfa, Brassica, and the grass Festuca under different planting densities. The results suggest that plants maintain their microbiome composition regardless of competition or the density of neighboring plants. I believe the paper is well-structured and provides valuable insights into the interactions between plants and their rhizosphere microbiomes.

I have a few comments and suggestions that I hope will help improve the manuscript:

Introduction

Page 10, line 69: It seems that a word ("that") is missing in this sentence. Please review for clarity.

Page 10, line 70: The term “partial disinfected” is unclear. Could the authors clarify what this means in this context?

Methods

Density and Diversity Greenhouse Experiments & Rhizosphere Soil Collection:

To improve clarity and aid in understanding the experimental design, I strongly recommend adding a schematic diagram that visually represents the experimental setup. This would be helpful for readers to better grasp the methodology.

Results

Page 14, line 328: In Fig. 5a, there is a red line indicating a negative correlation between Bacillus sp. S3 and Flavisolibacter ginsengiterrae. It would be important to mention this correlation in the results section, particularly in the part discussing the Brassica network analysis.

Discussion

Page 19, lines 450-452: I appreciate that the authors have included the potential allelopathic effects of Alfalfa on its microbiome. This is a crucial aspect of plant-microbe interactions that is often overlooked in ecological studies, and it adds an important layer of insight to the paper.

Conclusions

Page 30, line 538: It would be beneficial to briefly discuss the implications of mono- vs polyculture systems with respect to their effects on rhizosphere microbiomes. This could provide valuable context for how plant diversity might influence microbial communities.

Overall, I find this to be a well-conducted and important study that contributes to our understanding of plant-microbe interactions, particularly under varying plant densities. With the above revisions, I believe the manuscript will be even clearer and more impactful.

**Reviewer #2: ** Review remarks on the research article "Effect of intra- and inter-specific plant interactions on the rhizosphere microbiome of a single target plant at different densities" submitted by Newberger et al., to PLOS ONE. Authors discussed the rhizosphere plants microcosm study containing different combinations and densities (1-3 plants, 24 plants, and 48 plants) of Medicago sativa, Brassica sp., and Fescue sp. plants. Interestingly, plant density had a significant influence over beta diversity while plant diversity was found to be a less important factor since it did not have a significant change.

The present revised article can be acceptable in it present form. No further corrections are required.

6. PLOS authors have the option to publish the peer review history of their article (what does this mean? ). If published, this will include your full peer review and any attached files.

**Do you want your identity to be public for this peer review?** For information about this choice, including consent withdrawal, please see our Privacy Policy .

Reviewer #1: No

Reviewer #2: **Yes: ** Krishan K. Verma

---

## [Author Response · Author response to Decision Letter 1]

22 Nov 2024

Response to Reviewers for PLOS ONE Decision: Revision required [PONE-D-24-42703] - Effect of intra- and inter-specific plant interactions on the rhizosphere microbiome of a single target plant at different densities [EMID:d6bdd859297b6ec4]

We thank the editor of PLOS ONE and the reviewers for taking the time to review the revised manuscript. Below we respond to each point raised concerning this manuscript.

Journal Requirements:

If applicable, we recommend that you deposit your laboratory protocols in protocols.io to enhance the reproducibility of your results. Protocols.io assigns your protocol its own identifier (DOI) so that it can be cited independently in the future.

Three protocols were added to the manuscript. The DOI was added for two protocols, the “Cost-effective, high-throughput DNA sequencing libraries for multiplexed target capture”, and the unmodified “Library preparation protocol to sequence full length 16S rRNA gene in Nanopore MinION sequencer”. The modified protocol, “Library preparation and sequencing for MinION – Manter Lab Protocol” was added as a Supporting Information during the manuscript submission process.

Thank you for adding the hyperlinks for the PLOS ONE's style requirements for manuscripts. For the resubmission, we have looked closely at both documents, main body and author affiliations, and made the necessary changes. All headings are sentence case, supplementary files are now separated and appropriately named, affiliations are now in number only, title is centered, Zip code have been removed, abbreviations have been removed, contribution has been specified.

2. As noted, this manuscript relates to your previous publication in Scientific Reports. In your manuscript, please ensure you cite, discuss, and acknowledge overlap with the related work, and provide adequate justification (in the Introduction) for the new submission in light of the related published work.

We have cited, discussed, and acknowledge overlap with the related work which looked at bulk soils under cover crops of increasing densities and diversities. We have provided additional details and justification in the introduction for the new submission in light of the conclusions made for the new submission.

This original research article was financially supported by National Institute of Food and Agriculture (NIFA)/ United States Department of Agriculture (USDA) through a Western Sustainable Agriculture Research and Education (SARE) project #SW20-910. Funding was awarded to J.M. and D.M., and the sponsors did not play any role in the study design, data collection and analysis, decision to publish, or preparation of the manuscript.

https://www.nifa.usda.gov/

https://western.sare.org/

Please provide an amended statement that declares *all* the funding or sources of support (whether external or internal to your organization) received during this study, as detailed online in our guide for authors at http://journals.plos.org/plosone/s/submit-now . Please also include the statement “There was no additional external funding received for this study.” in your updated Funding Statement.

Thank you for clarifying that our Funding Statement was identified, clearly stated details which were additionally required, and for communicating that it will need to be placed in the cover letter so that the editor may change the submission on our behalf. We have re-read the PLOS ONE guidelines for the Funding Statement (we found that this link to be more helpful than the provided link above https://journals.plos.org/plosone/s/submission-guidelines#loc-additional-information-requested-at-submission ). Furthermore, we have added the term “all” and the phrase “There was no additional external funding received for this study.” so that the statement which is in the updated manuscript and the cover letter now reads as:

All funding for this original research article came from the National Institute of Food and Agriculture (NIFA)/ United States Department of Agriculture (USDA) through a Western Sustainable Agriculture Research and Education (SARE) project #SW20-910. Funding was awarded to J.M. and D.M. The funders had no role in study design, data collection and analysis, decision to publish, or preparation of the manuscript. There was no additional external funding received for this study.

https://www.nifa.usda.gov/

https://western.sare.org/

We thank the editor to emphasize the importance of making the data available before acceptance. Laboratory protocol for DNA sequencing, RStudio files for analysis and data visualization, relative abundance sequence counts, metadata file, and a README file are all on GitHub on https://github.com/Derek-Newberger/Plant_Neighbor_Rhizosphere.git and on [https://github.com/DanielManter-USDA/DRN-2381389].

5. Please upload a new copy of Figures 3, 5 and 7 as the detail is not clear. Please follow the link for more information: https://blogs.plos.org/plos/2019/06/looking-good-tips-for-creating-your-plos-figures-graphics/"
https://blogs.plos.org/plos/2019/06/looking-good-tips-for-creating-your-plos-figures-graphics/"

We thank the editor for providing the helpful link. To maintain the high quality for the figure, we have uploaded the exported file from Gephi. Although in this way, each figure is broken down to the a, b, c, and d parts for figures 3, 5, and 7, the quality for each figure is maintained.

6. Please review your reference list to ensure that it is complete and correct. If you have cited papers that have been retracted, please include the rationale for doing so in the manuscript text or remove these references and replace them with relevant current references. Any changes to the reference list should be mentioned in the rebuttal letter that accompanies your revised manuscript. If you need to cite a retracted article, indicate the article’s retracted status in the References list and also include a citation and full reference for the retraction notice.

We have reviewed the citations and manually looked each one up and can confirm that none of them have been retracted. Additionally, in text citations now use a hyphen if they are in consecutive order of more than three references together and come in numerical order.

Reviewers' comments to the author:

Reviewer #1: This is an interesting study where the authors examine, through a pot microcosm experiment, the rhizosphere microbiome of Alfalfa, Brassica, and the grass Festuca under different planting densities. The results suggest that plants maintain their microbiome composition regardless of competition or the density of neighboring plants. I believe the paper is well-structured and provides valuable insights into the interactions between plants and their rhizosphere microbiomes.

We thank Reviewer #1 in taking the time to read our resubmission and providing the insights in order to bring this article to be well-structured and provides valuable insights on the topic of rhizosphere microbiomes.

I have a few comments and suggestions that I hope will help improve the manuscript:

Introduction

Page 10, line 69: It seems that a word ("that") is missing in this sentence. Please review for clarity.

We have reviewed line 69 and added the word “that”.

Page 10, line 70: The term “partial disinfected” is unclear. Could the authors clarify what this means in this context?

We have clarified the term “partial disinfected” and changed it to “disinfected”. Previously we had it as “partially sterilized” since we recognized that while soil microbial loads were lowered they were not completely eliminated and therefore the term “disinfected” was more accurate and clear.

Methods

Density and Diversity Greenhouse Experiments & Rhizosphere Soil Collection:

To improve clarity and aid in understanding the experimental design, I strongly recommend adding a schematic diagram that visually represents the experimental setup. This would be helpful for readers to better grasp the methodology.

We agree with Reviewer #1 that the experimental design needed clarity. We have added supplementary table 1 that visually represents the experimental setup of the pots with increasing plant densities and diversities.

Results

Page 14, line 328: In Fig. 5a, there is a red line indicating a negative correlation between Bacillus sp. S3 and Flavisolibacter ginsengiterrae. It would be important to mention this correlation in the results section, particularly in the part discussing the Brassica network analysis.

We have mentioned the negative correlation between Bacillus sp. S3 and Flavisolibacter ginsengiterrae as it is interesting since all other correlations were positive.

Discussion

Page 19, lines 450-452: I appreciate that the authors have included the potential allelopathic effects of Alfalfa on its microbiome. This is a crucial aspect of plant-microbe interactions that is often overlooked in ecological studies, and it adds an important layer of insight to the paper.

We thank Reviewer #1 for appreciating the line that highlights the potential allelopathic effects of alfalfa on its microbiome.

Conclusions

Page 30, line 538: It would be beneficial to briefly discuss the implications of mono- vs polyculture systems with respect to their effects on rhizosphere microbiomes. This could provide valuable context for how plant diversity might influence microbial communities.

In the conclusion section, we have emphasized the implications of mono- vs polyculture systems with respect to their effects on rhizosphere microbiomes.

Overall, I find this to be a well-conducted and important study that contributes to our understanding of plant-microbe interactions, particularly under varying plant densities. With the above revisions, I believe the manuscript will be even clearer and more impactful.

Reviewer #2: Review remarks on the research article "Effect of intra- and inter-specific plant interactions on the rhizosphere microbiome of a single target plant at different densities" submitted by Newberger et al., to PLOS ONE. Authors discussed the rhizosphere plants microcosm study containing different combinations and densities (1-3 plants, 24 plants, and 48 plants) of Medicago sativa, Brassica sp., and Fescue sp. plants. Interestingly, plant density had a significant influence over beta diversity while plant diversity was found to be a less important factor since it did not have a significant change.

The present revised article can be acceptable in its present form. No further corrections are required.

We thank Reviewer #2 for reviewing the revised manuscript and greatly appreciate that the revisions were found to be satisfactory.

---

## [Decision Letter · Decision Letter 1]

16 Dec 2024

Effect of intra- and inter-specific plant interactions on the rhizosphere microbiome of a single target plant at different densities

PONE-D-24-42703R1

Dear Dr. Vivanco,

We’re pleased to inform you that your manuscript has been judged scientifically suitable for publication and will be formally accepted for publication once it meets all outstanding technical requirements.

Kind regards,

Eugenio Llorens

Academic Editor

PLOS ONE

Reviewers' comments:

Reviewer's Responses to Questions

**Comments to the Author**

1. If the authors have adequately addressed your comments raised in a previous round of review and you feel that this manuscript is now acceptable for publication, you may indicate that here to bypass the “Comments to the Author” section, enter your conflict of interest statement in the “Confidential to Editor” section, and submit your "Accept" recommendation.

Reviewer #1: All comments have been addressed

2. Is the manuscript technically sound, and do the data support the conclusions?

Reviewer #1: Yes

3. Has the statistical analysis been performed appropriately and rigorously? 

Reviewer #1: Yes

4. Have the authors made all data underlying the findings in their manuscript fully available?

Reviewer #1: Yes

5. Is the manuscript presented in an intelligible fashion and written in standard English?

Reviewer #1: Yes

6. Review Comments to the Author

Reviewer #1: The revised article is now acceptable in its current form, with no further corrections required. The authors have addressed all the corrections and suggestions provided

7. PLOS authors have the option to publish the peer review history of their article (what does this mean? ). If published, this will include your full peer review and any attached files.

**Do you want your identity to be public for this peer review?** For information about this choice, including consent withdrawal, please see our Privacy Policy .

Reviewer #1: No

---

## [Editor Report · Acceptance letter]

PONE-D-24-42703R1

PLOS ONE

Dear Dr. Vivanco,

I'm pleased to inform you that your manuscript has been deemed suitable for publication in PLOS ONE. Congratulations! Your manuscript is now being handed over to our production team.

Kind regards,

on behalf of

Dr. Eugenio Llorens

Academic Editor

PLOS ONE